# Neuroimaging of headaches in patients with normal neurological examination: protocol for a systematic review

Bernold Kenteu,[1] Yannick F Fogang,[2] Ulrich Flore Nyaga,[3] Joseline G Zafack,[4] Jean Jacques Noubiap,[5] Joseph Kamtchum-Tatuene[6,7]

## ABSTRACT

**Introduction** Headache disorders (HD) are among the most frequent neurological disorders seen in neurology practice. Because secondary HD are rare, patients' examination is most often unremarkable. However, the will to relieve patients' anxiety and the fear of prosecutions lead to overuse of neuroimaging thus resulting in the discovery of incidental findings (IF) or normal variants that can lead to futile or harmful procedures. Knowing the probability of identifying a potentially clinically significant lesion in patients with isolated headache could facilitate decision-making and reduce health costs. This review aims to determine the prevalence of incidental findings and normal anatomic variants (NAV) on neuroimaging studies performed in patients presenting with headache and normal neurological examination.

**Method and analysis** Studies reporting neuroimaging findings in patients with headache and normal neurological examination and published before the 30 September 2017 will be identified by searching PubMed, Medline and EMBASE (Excerpta Medica Database). Relevant unpublished papers and conference proceedings will also be checked. Full texts of eligible studies will then be accessed and data extracted using a standard data extraction sheet. Studies will be assessed for quality and risk of bias. Heterogeneity of studies will be evaluated by the $\chi^2$ test on Cochrane's Q statistic. The prevalence of NAV and IF across studies and in relevant subgroups will be estimated by pooling the study-specific estimates using a random-effects meta-analysis. Visual analysis of funnel plot and Egger's test will be used to detect publication bias. The report of this systematic review will be compliant with the Meta-analysis of Observational Studies in Epidemiology guidelines.

**Ethics and dissemination** The current study is based on published data; ethical approval is, therefore, not required. The final report of this systematic review will be published in a peer-reviewed journal. Furthermore, findings will be presented at conferences and submitted to relevant health authorities.

**Trial registration number** CRD42017079714.

## Strengths and limitations of this study

► To the best of our knowledge, this will be the first systematic review reporting the prevalence of incidental findings and normal variants in patients with normal neurological examination undergoing neuroimaging for headache.

► We will use robust statistical tools to pool prevalence across studies and this will ensure the reliability of our estimates.

► A major limitation would be the heterogeneity between included studies in terms of availability of advanced neuroimaging equipment (CT and/or MRI), expertise of the clinician performing the neurological examination and the radiologist interpreting the scans, variability of the imaging protocols.

► Another possible limitation could be the insufficient description of the clinical features of headaches in the selected studies which would limit the scope of our subgroup analyses and our ability to provide practical recommendations for the selection of patients presenting with headache and normal neurological examination that deserve brain imaging.

For numbered affiliations see end of article.

**Correspondence to**
Dr Bernold Kenteu;
a.bernold@yahoo.fr

## INTRODUCTION

Headache disorders (HD) affect people of all ages and races worldwide. With an estimated global prevalence of 50%,[1] HD are among the most frequent neurological disorders seen in primary care setting and in neurology practice. The International Classification of Headache Disorders differentiates between primary headaches, which are disorders caused by independent pathomechanisms, and secondary headaches, which are symptomatic of another condition known to cause the pain. Primary headaches constitute by far the most represented type of HD, tension-type headache and migraine being the most frequent, with a prevalence of 60% and 15%, respectively.[2] Despite their benign character, HD are a global public health problem due to the disability and medication overuse they cause and also their cost to the society.[3 4] In England, migraine is responsible for a loss of 25 million days from work or school every year and is associated with an annual cost of about 17 billion dollars in the USA.[5 6] The diagnosis of headache is based on a thorough

history taking and a good physical examination seeking to exclude or confirm a secondary cause. Since the most common type of HD are primary headaches, the physical examination will generally be unremarkable and neuroimaging unnecessary.[7] In spite of the relative rarity of secondary HD, the complex presentation of HD frequently raises the fear of serious underlying causes and thus regularly confront physicians with the question of whether or not to perform neuroimaging. The family request, the relief of patient's anxiety and the fear of lawsuit are others reasons for prescribing neuroimaging. These concerns lead to an overuse of neuroimaging and to the frequent discovery of normal variants (NV) and incidentals findings (IF) which most often do not explain the patient's pain.[8–11]

IF are defined as apparently asymptomatic intracranial abnormalities that are clinically significant because of their potential to cause symptoms or influence treatment. They can be classified as vascular (silent brain infarct, lacunes, microbleeds, structural vascular abnormalities and white matter hyperintensities) or non-vascular lesions. The latter can be further divided into neoplastic lesions (benign and malignant tumours) and non-neoplastic lesions (cysts, inflammatory lesions, hydrocephalus, Arnold-Chiari malformations and extraaxial collections).[12] NV are defined as anatomical variants that do not have the potential to cause symptoms and do not need any therapeutic intervention (eg, large cisterna magna and ventricular asymmetry).[12]

Several studies conducted in different settings and using different methodological approaches have produced variable estimates of the prevalence of normal and abnormal brain imaging in patients presenting with headache and normal neurological examination. Because the discovery of an IF or a NV on a brain imaging can sometimes prompt more worries for the patient and lead to futile and even harmful surgical procedures, knowing the probability of identifying a potentially clinically significant lesion (subset of IF) in patients presenting with isolated headache could help to facilitate decision-making for clinicians and reduce healthcare costs by avoiding a number of unnecessary scans.

## REVIEW QUESTION

What is the prevalence of incidental findings and normal anatomic variants on neuroimaging studies performed in patients presenting with headache and normal neurological examination?

## METHODS

This review protocol has been prepared according to the 2015 Preferred Reporting Items for Systematic review and Meta-Analysis Protocols (PRISMA-P) guidelines.[13] A PRISMA-P checklist is provided as the online supplementary appendix 1. The protocol is registered in the PROSPERO International Prospective Register of systematic reviews

(registration number CRD42017079714). The proposed start date for this review is 15 December 2017 and the entire work is expected to be completed in a maximum of 6 months. The timeline for the review is provided as the online supplementary appendix 2.

## Criteria for considering studies for the review
### Inclusion criteria
All observational studies reporting neuroimaging findings in patients presenting with headache and normal neurological examination will be included without date or language restriction.

### Exclusion criteria
► Case series with small sample sizes (<30 subjects)
► Studies lacking data to compute prevalence and/or explicit method description.
► Duplicates (for studies leading to more than one publication, only the most comprehensive report including the largest sample size will be considered).
► Studies whose full data will not be accessible even after request from authors.

## Search strategy for identifying relevant studies
The research strategy will be implemented in two stages.

### Bibliographic database searches
A comprehensive and exhaustive search on PubMed, MEDLINE and EMBASE (Excerpta Medica Database) will be conducted to identify all relevant articles reporting neuroimaging findings in patients presenting with headache and normal neurological examination and published before the 30 September 2017. Both plain language words and medical subheadings (MeSH) will be used. Abstracts of all eligible papers will be reviewed, and full texts of articles will be accessed through PubMed, Google Scholar, HINARI or journals' websites. The detailed search strategy for PubMed and EMBASE are shown in table 1 and 2, respectively.

### Searching for other sources
References of all relevant original and review articles will be scrutinised for potential additional data sources, and their full texts will be accessed in a similar way. Conference proceedings will also be checked to identify relevant unpublished data. In case some full-text papers are not accessible via the internet-based sources, authors will be contacted by email to provide reprints and/or related data. All sources of additional data will be documented and clearly referenced in order to allow verification if necessary.

| Table 1 | Search strategy for PubMed |
|---|---|
| #1 | 'headache*' |
| #2 | 'neuroimaging' OR 'brain imaging' OR 'CT scan' OR 'MRI scan' |
| #3 | #1AND #2 |
| #4 | Restrict [humans] |

**Table 2** Search strategy for EMBASE

| Database | | Search strategy |
|---|---|---|
| Embase | #1 | 'headache*':ti,ab OR 'cephalgia*':ti,ab OR 'cephalalgia*':ti,ab OR 'cranialgia*':ti,ab OR 'head ache*':ti,ab OR 'cephalodynia*':ti,ab OR 'cephalea*':ti,ab OR 'cerebral pain':ti,ab OR 'head pain':ti,ab OR 'eye pain':ti,ab |
| | #2 | 'neuroimaging':ti,ab OR 'brain imaging':ti,ab OR 'tomography':ti,ab OR 'mri':ti,ab OR 'magnetic resonance imaging':ti,ab OR 'mr imaging':ti,ab OR 'nmr imaging':ti,ab |
| | #3 | #1AND #2 |
| Restrict to humans | #4 | #3 AND 'human'/de |
| Filter by type of study | #5 | #4 AND ('clinical study'/de OR 'clinical trial'/de OR 'cohort analysis'/de OR 'comparative study'/ de OR 'controlled clinical trial'/de OR 'controlled study'/de OR 'family study'/de OR 'major clinical study'/de OR 'medical record review'/de OR 'observational study'/de OR 'prospective study'/de OR 'randomized controlled trial'/de OR 'retrospective study'/de OR 'systematic review'/de) |

### Selection of studies for inclusion in the review

Titles and abstracts of records identified through literature search will be independently screened for eligibility by two members of the research team (BK and JKT). Full texts of studies deemed eligible will be retrieved and further assessed for inclusion by the same investigators. Any disagreement will be resolved by discussion and consensus. If the latter is not reached, arbitration will be sought from a third member of the team (YFF). The inte-rater agreement for the selection of studies will be assessed using a non-weighted Cohen's kappa statistic.[14 15]

### Assessment of methodological quality and data reporting

Two independent assessors (JKT and JJN) will use the Risk of Bias Tool for Prevalence Studies (online supplementary appendix 3)[16] to evaluate the methodological quality and risk of bias for each study using the full-text publication. To each item, they will assign a score of 1 (yes) or 0 (no), and will sum scores across items to generate an overall quality score ranging from 0 to 10. According to the overall scores, we will classify studies as having a low (>8), moderate (6–8) or high (≤5) risk of bias. Risk of bias scores will be presented in a table and inter-rater agreement will be assessed using a weighted Cohen's kappa statistic.[17 18]

### Data extraction and management

Search results will be compiled using the citation management software, EndNote X7.2.1. A data extraction sheet will be used to collect the following information:

► General information: first author name, year of publication, year of participants' inclusion, country, type of publication and language of publication (full text).
► Study characteristics: study design, setting (hospital, population and emergency department), sample size, mean or median age, age range, proportions of male participants, proportion of acute versus chronic versus recurrent headache, type of neuroimaging used (CT or MRI, without and/or with contrast), power of the MRI magnetic field (0.35, 0.5, 1.5 or 3 Tesla), qualification or the person reading the images (radiologist and neuroradiologist), qualification of the person doing the clinical assessment (general practitioner,

emergency physician and neurologist), proportion of HIV positive, proportion of patients with fever, proportion of patients with history of head trauma, criteria used for the clinical diagnosis and classification of headache, and proportion of migraines.
► Neuroimaging findings in patients with normal neurological examination.

### Data synthesis including assessment of heterogeneity

Data will be analysed using the software STATA (V.13). Inter-rater agreement for study inclusion and data extraction will be assessed using Cohen's kappa ($\kappa$) coefficient.[18] Study-specific estimates will be determined from the point estimate and the appropriate denominators, assuming a binominal distribution. Then, the study-specific estimates will be pooled through a random-effects meta-analysis to obtain an overall summary estimate of the prevalence across studies, after stabilising the variance of individual studies using the Freeman-Tukey double arc-sine transformation.[19] Heterogeneity will be evaluated by the $\chi^2$ test on Cochrane's Q statistic which is quantified by $I^2$ values, assuming that $I^2$ values of 25%, 50% and 75%, represent low, medium and high heterogeneity, respectively.[20] Where substantial heterogeneity will be detected, a subgroup analysis will be performed to detect its possible sources. Visual analysis of funnel plot and Egger's test will be done to detect publication bias.[21] All tests will be two-sided and statistical significance will be defined as $P < 0.05$.

### Results reporting and presentation

The resulting systematic review and meta-analysis will follow the Meta-analysis of Observational Studies in Epidemiology guidelines for reporting.[22] The study selection process will be summarised using a flow diagram. Reasons for study exclusion will be described. Quantitative data will be presented in summary tables and forest plots where appropriate. The quality scores and risk of bias for each eligible study will be reported.

### Ethics and dissemination

This systematic review and meta-analysis will be based on data from ethically approved studies. Therefore, ethical

approval is not required. The final report of this study, in the form of a scientific paper, will be published in peer-reviewed journals. Findings will be further presented at conferences and submitted to relevant health authorities. We also plan to monitor publications on the topic and to update the review accordingly .

**Author affiliations**
[1]Service de Radiologie et Imagerie Médicale, Département de Biologie et Explorations Fonctionnelles, Faculté de Médecine, de Pharmacie et d'Ontostomatologie, Université de Cheikh Anta Diop, Dakar, Senegal
[2]Head Neuropsychiatry Department, Bafoussam Regional Hospital, Bafoussam, Cameroon
[3]Department of Internal Medicine and Specialties, Faculty of Medicine and Biomedical Sciences, University of Yaoundé I, Yaounde, Cameroon
[4]Department of Social and Preventive Medicine, Laval University, Quebec, Canada
[5]Department of Medicine, Groote Schuur Hospital and University of Cape Town, Cape Town, South Africa
[6]Brain Infections Group, Institute of Infection and Global Health, University of Liverpool, Liverpool, UK
[7]Malawi-Liverpool-Wellcome Trust Clinical Research Programme, Blantyre, Malawi

**Contributors** BK, UFN, JJN and JK-T: study conception. BK and UFN: manuscript drafting. YFF, JGZ, JJN and JK-T: critical revision of manuscript. BK, YFF, UFN, JGZ, JJN and JK-T: final approval of the version to be published. JK-T: guarantor of the review.

**Funding** This research received no specific grant from any funding agency in the public, commercial or not-for-profit sectors.

**Competing interests** None declared.

**Patient consent** Not required

**Provenance and peer review** Not commissioned; externally peer reviewed.

**Data sharing statement** There is no additional unpublished date from this study available.

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
