## [Reviewer comments · BMJ Open]

ARTICLE DETAILS

TITLE (PROVISIONAL)	Neuroimaging of headaches in patients with normal neurologic examination: protocol for a systematic review
AUTHORS	KENTEU, Averik Bernold; Fogang, Yannick; Nyaga, Ulrich Flore; Zafack, Joseline; Noubiap, Jean Jacques; Tatuene, Joseph

VERSION 1 – REVIEW

REVIEWER	Nadja Kadom, MD Emory University, Atlanta, Georgia, USA My research area is in pediatric headaches imaging but I am not currently planning a meta-analysis
REVIEW RETURNED	30-Oct-2017

GENERAL COMMENTS	General: The idea for the review is good. While publication of protocols for prospective trials is common, it is very uncommon to do so for a systematic review. There is no compelling argument made for how the knowledge of incidental findings in headache imaging can inform imaging practices. The methods section is severely lacking in detail. Title: Appropriate Abstract: Introduction: • The introduction does not explain how knowing more about incidental findings and normal variants relates to the relevance of neuroimaging. Per definition, incidental findings and normal variants would be irrelevant to the diagnosis of headaches. Instead, as the authors state themselves, the concern is for serious underlying causes. It is unclear why the authors are not focusing on severe underlying causes.• The authors did not provide enough information on the study question to deduce that a PICO(S) statement exists Methods and Analysis: • The authors do not state the exact start date, they list only the end date for the search.• Study eligibility, inclusion/exclusion criteria are not listed• The authors need to state the source of the unpublished papers and conference proceedings and how these could be verified.• The authors need to state whether all language articles will be considered and translated.• The study appraisal criteria are not stated• The study synthesis method is not described
---

	 • The authors do not state whether they registered this systematic review protocol with PROSPERO and that they searched whether a similar search protocol already exists Ethics: The authors state the study is based on published data, but in the Methods section it is stated that unpublished work will also be included Introduction:  • The authors did not provide enough information on the study question to deduce all components of the PICO(S) statement. The population appears to be patients with primary headaches; the intervention seems to be neuroimaging (CT? MRI? Xrays? ALL?); what is the comparison?, what are the outcomes (e.g. what exactly will be considered a “risk” and a “benefit”).  Methods:  • The authors are not selecting a start date. This is problematic, as the authors refer to the ability for clinicians to distinguish between primary and secondary headaches based on the clinical exam. The first headache classification by the ICHD was published ~1988. CT imaging was not available before ~1975, and MRI became available around ~1994. • Search strategy; the authors are only using the term “headache”, but commonly the indication will state “migraine” or could state “head pain”, “eye pain” or other terms. It is unclear how the authors justify using only the term headaches • In search #1 the author are proposing to us headaches “OR” other terms, when it should be “AND” other terms • The authors do not explain how they arrived at the list of search terms in search #2. • Assessment of quality: The authors cannot simply refer to the PRISMA checklist, they need to detail exactly what kind of bias will be scored and how and who and qualification for scoring of those who will score • Data extraction: It is not clear from the description how the authors will define a primary headache from the studies selected • The authors need to provide a list of what they will consider an incidental finding or normal variant, along with a rationale why these findings are unrelated to headaches 
--	--

REVIEWER	Fayyaz Ahmed Hull York Medical School Hull Royal Infirmary Anlaby Road Hull HU3 2JZ
REVIEW RETURNED	10-Nov-2017

GENERAL COMMENTS	I suggest the authors elaborate on the limitations for the review a little further than just the heterogeneity of the studies. This is an area where expertise of the clinician, centre and publisher may come up with different conclusions. In addition it will be a good idea to define which anatomical variants would be particularly looked for.
--

VERSION 1 – AUTHOR RESPONSE

Reviewer 1:

Abstract

The comments on the abstract have been repeated in the series of comments related to the main text of the protocol. As a general response, we would say that the word restriction for the abstract limits the amount of information that can be inserted. Details regarding each aspect of the protocol are provided in the main text. For detailed answers to all the queries, we kindly invite the Reviewer to read below.

Introduction

1. The introduction does not explain how knowing more about incidental findings and normal variants relates to the relevance of neuroimaging. Per definition, incidental findings and normal variants would be irrelevant to the diagnosis of headaches. Instead, as the authors state themselves, the concern is for serious underlying causes. It is unclear why the authors are not focusing on severe underlying causes.

Response:

We thank the Reviewer for prompting us to clarify our definition of the expressions “Incidental Findings” and “Normal Variant”. We have designed our protocol using the definitions provided by Morris et al, BMJ 2009 (PMID 19687093). These definitions have been inserted in the introduction (see page 5 and 6, lines 95 – 103).

IF are defined as apparently asymptomatic intracranial abnormalities that were are clinically significant because of their potential to cause symptoms or influence treatment. They can be classified as vascular (silent brain infarct, lacunes, microbleeds, structural vascular abnormalities, white matter hyperintensities) or non-vascular lesions. The latter can be further divided into neoplastic lesions (benign and malignant tumors), non-neoplastic lesions (cysts, inflammatory lesions, hydrocephalus, Arnold-Chiari malformations, and extra-axial collections). NV are defined as anatomical variants that do not have the potential to cause symptoms and do not need any therapeutic intervention (e.g. large cisterna magna, ventricular asymmetry).

Taking these definitions into account, it appears that our work will not only satisfy the Reviewer’s interest about serious underlying pathologies that could have been missed had brain imaging not been done but will also go beyond to provide useful epidemiologic/descriptive information on the prevalence of IF and NV in patients with headache and normal neurologic examination.

2. The authors did not provide enough information on the study question to deduce all components of the PICO(S) statement. The population appears to be patients with primary headaches; the intervention seems to be neuroimaging (CT? MRI? Xrays? ALL?); what is the comparison? what are the outcomes (e.g. what exactly will be considered a “risk” and a “benefit”).

Response:

We thank the Reviewer for giving us this opportunity to clarify the nature of our research. We are conducting a systematic review of observational (not interventional) studies with the goal of summarizing the prevalence of IF and NV (finding/outcome) on neuroimaging studies (diagnostic procedure/intervention) in patients with headache and normal neurologic examination (population). Ideally, the comparison would have been made with healthy volunteers (no headache and normal neurologic examination) but the latter do not undergo systematic neuroimaging. A comparison is therefore not applicable for this review. The PICO(S) statement is not applicable to all types of research questions.

Methods

3. The authors are not selecting a start date. This is problematic, as the authors refer to the ability for clinicians to distinguish between primary and secondary headaches based on the clinical exam. The first headache classification by the ICHD was published ~1988. CT imaging was not available before ~1975, and MRI became available around ~1994.

Response:

We thank the Reviewer for these very useful historical milestones. However, given that our review is clearly focused on neuroimaging, we do not think that indicating a start date will change the yield of our database searches. It is common practice to search databases from inception (1950 for PubMed, 2008 for EMBASE). We do not expect to find any article reporting brain imaging findings before relevant equipment (CT or MRI) became available in routine clinical practice. Moreover, in the paragraph describing data extraction (see page 9, lines 174 – 189), we have indicated that we will document the year of publication and the type of brain imaging technique used for each study. This information will be used in subgroup analyses to clarify how the date of publication and the type of neuroimaging technique used influence the prevalence of IF and NV.

4. Study eligibility, inclusion/exclusion criteria are not listed

Response:

As mentioned above, the eligibility (inclusion/exclusion) criteria for studies were not listed in the abstract because of the word restriction, but are listed in the manuscript (see page 6, lines 126 – 135).

5. The authors need to state the source of the unpublished papers and conference proceedings and how these could be verified.

Response:

We thank the Reviewer for this query. As mentioned above, some information were not placed in the abstract because of the word restriction. But the related paragraph in our manuscript reads as follows “References of all relevant original and review articles will be scrutinized for potential additional data sources, and their full texts will be accessed in a similar way. Conference proceedings will also be checked to identify relevant unpublished data.”

The expression “unpublished papers” classically refers to data that were not included in a citable journal article. It is not possible to identify a priori which specific article will refer to unpublished data or which specific conference proceedings will contain the unpublished data of interest.

However, sources of additional data can be documented and referenced a posteriori to allow verification. This is now clearly stated in the manuscript (see page 8, lines 154 – 155).

6. The authors need to state whether all language articles will be considered and translated.

Response:

Unfortunately, because of the word restriction we were unable to insert a huge quantity of information in the abstract. But, in the section describing the inclusion criteria of our main text, we have indicated that papers will be considered without language restriction (see page 7, lines 128 – 129). We have colleagues in our network that could help with the translation whenever necessary.

7. The study appraisal criteria are not stated.

Response:

The study appraisal criteria were not detailed in the abstract because of the word restriction. We thank the Reviewer for this concern which has allowed us to clarify the study appraisal criteria. We have reformatted the related paragraph in our manuscript (see page 8 and 9, lines 165 – 172). It now reads as follows:

“Two independent assessors (JKT and JJN) will use the Risk of Bias Tool for Prevalence Studies (supplementary appendix 3) to evaluate the methodological quality and risk of bias for each study using the full-text publication. To each item, they will assign a score of 1 (yes) or 0 (no), and will sum scores across items to generate an overall quality score ranging from 0 to 10. According to the overall scores, we will classify studies as having a low (>8), moderate (6–8), or high (≤5) risk of bias. Risk of bias scores will be presented in a table and interrater agreement will be assessed using a weighted Cohen’s kappa statistic.”

The reader of the protocol can access the supplementary appendix to see which parameters will be assessed.

8. The study synthesis method is not described

Response:

We thank the Reviewer for this comment. Indeed the study synthesis method was not detailed in the abstract because of the word restriction. We have dedicated one paragraph of the protocol to describe the method for study synthesis (see page 10, lines 192 – 205).

9. The authors do not state whether they registered this systematic review protocol with PROSPERO and that they searched whether a similar search protocol already exists.

Response:

- At the time of submission the protocol had been submitted to the Centre for Reviews and Dissemination but we did not yet have a registration number. Since it’s now available, we added it at the beginning of the Method section (see page 6, lines 118 – 122).

- Checking that there is no similar work being done is a prerequisite for the registration process in PROSPERO.

10. Search strategy; the authors are only using the term “headache”, but commonly the indication will state “migraine” or could state “head pain”, “eye pain” or other terms. It is unclear how the authors justify using only the term headaches

Response:

We thank the Reviewer for prompting us to clarify our search strategy. Indeed, when designing a search strategy, one should aim to maximize the sensitivity (increase the maximum number of articles relevant to the topic that are retrieved) and specificity (limit the inclusion of irrelevant publications) but should also take into account how the databases are constructed and how the search engines operate.

EMBASE uses a collection of terms called Emtree in which each term is related to its synonyms in both scientific and plain language. PubMed used MeSH terms and keywords that are also systematically related to relevant synonyms. Therefore, searching for “Headache” in both databases and specifying that the term should be “exploded” (PubMed) or “expanded” (EMBASE) would retrieve all the articles that contain synonyms of headache, including those referred to by the Reviewer.

In the revised version of the manuscript, we have now provided updated search strategies for PubMed and EMBASE. The search terms suggested by the Reviewer were taken into consideration in the updated search strategies. These search strategies have been run and cross-checked a couple of times to make sure that they are accurate.

11. In search #1 the author are proposing to us headaches “OR” other terms, when it should be “AND” other terms

Response:

We do not agree with this statement. When building a search strategy for a given research question, the following principles apply:

- First, the question is broken down into a number of searches, the latter corresponding to the elements of the PICO(S) statement whenever possible.
- Second, each search (e.g. #1, #2, #3) is built using one element of the PICO(S) statement and its synonyms related with the Boolean operator “OR” rather than “AND” because the synonyms do not always appear together in a given publication. Using the Boolean operator “OR” maximizes the sensitivity of each search at the initial steps of the process.
- Third, the searches are combined using the Boolean operator “AND” in order to maximize the specificity i.e. to exclude all publications that are not relevant because at least one of the required predefined elements of the PICO(S) statement is missing.
- Finally, the filters are applied (e.g. human/animal studies, language restrictions, date restrictions).

12. The authors do not explain how they arrived at the list of search terms in search #2.

Response:

We thank the Reviewer for prompting us to clarify the content of search #2. Indeed, we used the article by Evans et al, Headache 2017 (PMID 28294311) to build a list of the various normal variants and incidental findings that could be found on neuroimaging in patients presenting with headache. However, after cross-checking the performance of our initial search strategy in PubMed and EMBASE, we realized that it was not necessary to include a list of the actual findings. A simple indication of the neuroimaging techniques was sufficient. The updated search strategies are provided in Tables 1 and 2.

13. Assessment of quality: The authors cannot simply refer to the PRISMA checklist, they need to detail exactly what kind of bias will be scored and how and who and qualification for scoring of those who will score.

Response:

- The paragraph describing the appraisal of the quality of studies has been rewritten to indicate all the information requested by the Reviewer. We will not use the PRISMA checklist but rather the validated Risk of Bias Tool for Prevalence Studies that has been (PMID 22742910). For details, please refer to the answer to question 8 above and to the main text of the manuscript (see page 8 and page 9, lines 165 – 172).
- The scorers (JKT and JJN) have received appropriate training for such task and have applied them consistently in previous systematic reviews.

14. Data extraction: It is not clear from the description how the authors will define a primary headache from the studies selected.

Response:

- We thank the Reviewer for giving us the opportunity to clarify the context of our work. We are not going to conduct a new study involving recruitment of new participants. In this systematic review, we will extract data from studies whose participants have been selected according to predefined clinical criteria (that are therefore beyond our control). However, because the selection criteria may vary from one study to the other, we have planned to document the “criteria used for the clinical diagnosis and classification of headache” (see page 9, line 187). This will help us to assess how the criteria used influence the prevalence of IF and NV on neuroimaging in each study (subgroup analyses).

- We would also want to mention that the scope of our review is not limited to primary headaches but to all cases of headache with normal neurologic examination. This means that patients with primary headache and abnormal neurologic examination will be excluded during data extraction because there is usually no debate on whether or not they should benefit from neuroimaging if affordable. The aim of the review is to assess the relevance of neuroimaging for patient presenting with headache and normal neurologic examination. We expect that at the end of the various subgroup analyses, we will be able to provide some practical recommendations to guide the selection of patients with headache that deserve brain imaging despite having a normal neurologic examination.

15. The authors need to provide a list of what they will consider an incidental finding or normal variant, along with a rationale why these findings are unrelated to headaches.

Response:

- Following the Reviewer's first question, we have included the definition of IF and NV in the introduction of our protocol (see page 5 and 6, lines 95 – 103). We have also cited three major publications (PMIDs: 28294311, 28984355, and 19687093) providing a list of what is considered as IF or NV by the various specialists caring for patients with headache. Those articles will be used as a reference when performing data extraction.

Ethics

16. The authors state the study is based on published data, but in the Methods section it is stated that unpublished work will also be included.

Response:

We thank the Reviewer for pointing out the mistake. We wanted to indicate that the review will be based on results of ethically approved studies, meaning that no further ethical approval will be required. We have revised the manuscript to reflect this (see page 10, lines 215).

Reviewer 2

1. I suggest the authors elaborate on the limitations for the review a little further than just the heterogeneity of the studies. This is an area where expertise of the clinician, centre and publisher may come up with different conclusions.

Response:

Following the Editorial Team's instructions (see editorial requirement N°3 at the beginning of this letter), we have deleted the conclusion discussing the limitations of our review. They are now discussed in the "Strengths and limitations section" right after the abstract. The Reviewer's suggestion has been taken into account.

2. In addition it will be a good idea to define which anatomical variants would be particularly looked for.

Response:

In the manuscript, we have cited three major publications (PMIDs: 28294311, 28984355, and 19687093) providing a list of what is considered as IF or NV by the various specialists caring for patients with headache. Those articles will be used as a reference when performing data extraction.

We are very grateful to the Reviewers and the Editorial Staff for their comments that helped us to improve the manuscript.

We hope that the current version now meets the requirements for acceptance and look forward to hearing from you.